# TVL: Policy Gradient without Bootstrapping via Truncated Value Learning

## Abstract

Reinforcement learning algorithms have typically used discounting to reduce the variance of return estimates. However, this reward transformation causes the agent to optimize an objective other than what is specified by the designer. We present a novel deep policy gradient algorithm, *Truncated Value Learning* (TVL), which can learn rewards *discount free* while simultaneously learning value estimates for *all* summable discount functions. Moreover, unlike many other algorithms, TVL learns values without bootstrapping. We hypothesize that bootstrap-free learning improves performance in high-noise environments due to reduced error propagation. We tested TVL empirically on the challenging high-noise *Procgen* benchmark and found it outperformed the previous best algorithm, Phasic Policy Gradient. We also show that our method produces state-of-the-art performance on the challenging long-horizon Atari game *Skiing* while using less than 1% of the training data of the previous best result.

## 1 Introduction

Discounting has been extensively used when implementing reinforcement learning (RL) algorithms. Theoretical concerns raise serious questions about the correctness of using this reward modification technique (Naik et al., 2019), especially with policy gradient algorithms (Nota & Thomas, 2019). The justification for discounting typically comes from the need to make returns finite (Sutton & Barto, 2018), which holds in the continuing setting but not for the more common time-limited episodic tasks. Despite the shortcomings, discounting is still used extensively on episodic tasks (Hessel et al., 2018; Mnih et al., 2015; Badia et al., 2020b;a) and with horizons much shorter than the typical duration of the episodes.

In this work, we argue that the primary need for discounting in deep RL is to dampen error propagation when bootstrapping. We demonstrate that a noisy function approximator, when combined with bootstrapping, can lead to divergence, even when used on policy.[1]. This implies that a deep RL algorithm that does not bootstrap might be more suitable for high-noise, long-horizon settings.

Fixed-horizon learning (De Asis et al., 2020) is notable in that it learns value estimates, not for a single horizon, but for each of a set of fixed horizons.[2] When performing updates, each value estimate uses fixed-horizon estimates strictly shorter than the horizon being estimated. For this reason, if the estimates are independent, no bootstrapping occurs. Fixed-horizon learning offers some interesting capabilities, including generating values estimates for arbitrary discount functions (see Figure 1), and can even generate discount-free estimates in the time-limited episodic setting.

Previously, fixed-horizon learning had been applied only to short-horizon tasks ($h_{\text{MAX}} = 64$) and using Q-learning (De Asis et al., 2020). However, this approach does not naturally align with continuous control problems or the learning of stochastic policies. In our research, we introduce fixed-horizon learning within policy gradient methods, extending its applicability to significantly longer horizons ($h_{\text{MAX}} = 27,000$). We achieve this by leveraging a new algorithm we call truncated value learning (TVL). We demonstrate that our approach produces state-of-the-art results on the high-noise

---

[1]Bootstrapping, function approximation, and off-policy learning are sometimes referred to as the deadly triad (Sutton & Barto, 2018). However, we show in Section 3 that divergence can occur even without being off-policy

[2]We refer to this as the 'value curve'

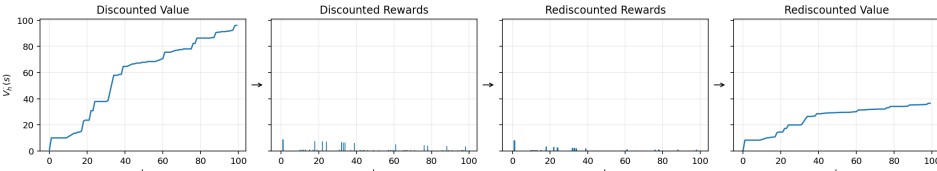

Figure 1: Typically, agents do not learn when rewards arrive, only their sum. However, in fixed-horizon learning, we learn a 'value curve' for all horizons up to $h_{\text{MAX}}$. By taking the finite difference over the curve, we recover the (expected) discounted rewards, which can then be rediscounted and summed together to give a value estimate for *any* discount function. Pictured here is the value curve under geometric discounting ($\gamma = 0.99$) being transformed into the value curve under the hyperbolic discount $1/(1 + h/10)$

.

*Procgen* environment and the long-horizon Atari game *Skiing* without compromising performance on traditional short-horizon Atari games.

The primary contributions of this work are:

- An efficient interpolation-based solution to enable efficient learning of very long fixed-horizon values.

- A sample-based return estimator that enables efficient TD($\lambda$)-style return estimates in the fixed-horizon setting.

- A process for using fixed-horizon estimates to generate advantage estimates suitable for policy gradient.

- The first results demonstrating fixed-horizon learning's effectiveness on long-horizon ($h \gg 100$) tasks.

We outline the background and relevant work in Section 2. Section Section 3 gives a simple example of how bootstrapping can fail on high-noise tasks. In Section 4, we give the details of our policy-gradient algorithm Truncated Value Learning (TVL). In Section 5 we evaluate TVL empirically on the high-noise *Procgen* environment and long-horizon Atari games.

## 2 BACKGROUND AND RELATED WORK

**MDPs.** We define an MDP $\mu$ as the tuple $(S, A, P, R, \gamma)$, where $S$ is the set of all states, $A$ the set of all actions, $P(s'|s, a)$ the probability of transitioning to state $s'$ when action $a$ is taken from state $s$, $R \ (S \times A \times S \mapsto \mathbb{R})$ the reward function, and $\gamma \in [0, 1)$ the discount factor. We also write a trajectory of state action pairs as $(s_0, a_0, s_1, a_1, ...)$ as $\tau$. At each time step $t$ the agent receives a state $s_t \in S$ and reward $r_t \in \mathbb{R}$, and must select an action $a_t \in A$ according to its policy $\pi(a|s)$.

It can also be useful to consider the discounted sum or rewards (return) from a state $s_t$ under some policy $\pi$ as

$$G^\gamma(\tau) := \sum_{k=0}^{\infty} \gamma^k r_{t+k}. \tag{1}$$

We can further define the value of being in state $s_t$ as the expected return

$$V^\gamma(s) := \mathbb{E}_{\tau \sim \pi, \mu}[G^\gamma(\tau)|s_0 = s], \tag{2}$$

as well as the $n$-step estimate

$$\text{NSTEP}^{(\gamma, k)}(\tau) := \left( \sum_{i=0}^{k-1} \gamma^i r_{t+i} \right) + \gamma^k V^\gamma(s_{t+k}). \tag{3}$$

Return estimates can be further improved by using an exponentially weighted sum of $n$-step returns (Sutton, 1988)

$$\text{TD}^{(\gamma,\lambda)}(\tau) := (1 - \lambda) \sum_{k=1}^{\infty} \lambda^{k-1} \text{NSTEP}^{(\gamma,k)}(\tau), \tag{4}$$

where $\lambda$ is a parameter that controls the variance / bias trade-off of the estimate.

**Fixed-horizon Learning.** Like traditional return estimators, a truncated return estimate is the expected sum of discounted future rewards. However, unlike a traditional return estimate, the rewards are ignored after some horizon $h$ (De Asis et al., 2020). Formally for some trajectory $\tau$, fixed horizon $h \in \mathbb{Z}^{\geq 0}$, and policy $\pi$, the finite-horizon discounted return is defined as

$$G_h^{\gamma}(\tau) := \sum_{t=0}^{h-1} \gamma^t r_t \tag{5}$$

with the expected value given by

$$V_h^{\gamma}(s) := \mathbb{E}_{\tau \sim \pi, \mu}[G_h^{\gamma}(\tau)|s_0 = s] \tag{6}$$
$$V_{\leq 0}(\cdot) := 0 \tag{7}$$

Estimates for $V_h^{\gamma}$ can be generated by return estimators with very similar properties to those used in traditional TD methods. Several return estimators are listed in Appendix C. However, due to the use of fixed-horizon discounting, TD($\lambda$) methods are inefficient to generate. For a more comprehensive review of fixed-horizon learning, see De Asis et al. (2020).

**Time-Limited Episodic Environments.** We limit our domain to time-limited episodic tasks. Formally, an episodic environment $\mu$ is time-limited if the terminal state $s_{\text{term}}$ will always be reached in at most $t_{\text{MAX}}$ time steps. We also require that the agent is given the current time step $t$ as part of the state space, as otherwise, time-limited environments would no longer be Markovian (Pardo et al., 2018).

Crucially, $t_{\text{MAX}}$ should not be misconstrued as a mere training detail but rather an essential property of the environment. That is, the goal is to maximize the accumulated reward in the time-limited environment, not in the time-unlimited environment. This is important as for many environments where the score can be accumulated indefinitely, the time-unlimited task is not well defined. Furthermore, the optimal policy for a small $t_{\text{MAX}}$ might differ considerably from that for a large $t_{\text{MAX}}$.

**Policy Gradient.** Our work builds on the widely adopted Proximal Policy Optimization (PPO) algorithm (Schulman et al., 2017), which employs a deep neural network (DNN) to learn a value function and a policy concurrently. Recognizing the intricacies of learning value curves, we adopt and expand upon the Dual-network architecture variant (PPO-DNA). This variant, characterized by its dual networks for policy and value, has demonstrated impressive results in vision-based tasks.

**Related Work.** Simultaneously learning multiple horizons was proposed by Fedus et al. (2019), who used the estimations to reconstruct an estimate of the hyperbolic discounted value. Their results showed no improvement for hyperbolic discounting but that learning this richer discounting value function was beneficial as an auxiliary task. Like our method, their method allows for learning arbitrary discount estimates. However, their method does not work in the undiscounted case $\gamma = 1$ and does not explicitly enable the estimates from one horizon to be used to update another.

The most similar work to ours is De Asis et al. (2020), which also uses fixed-horizon methods to learn a value function and provides a lot of the theoretical underpinning for this paper. Our work differs in two important ways. First, we apply fixed-interval learning to a Policy Gradient algorithm rather than Q-learning, which naturally supports stochastic policies and continuous control problems. Second, their method was significantly limited in the length of the horizons feasible ($h = 64$). In contrast, we develop methods to allow fixed-horizon to scale to much larger horizons ($h = 27,000$). Sun et al. (2018) also maximize the k-step rewards, but requires a cost-to-go oracle based on expert (but flawed) training.

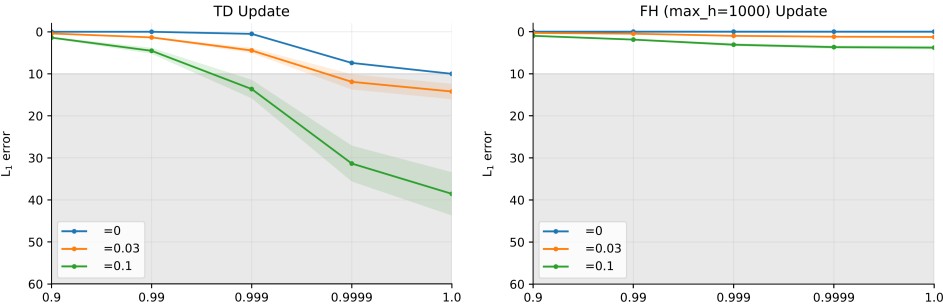

Figure 3: A toy example. On the left, we update the value estimates using standard TD updates. Undiscounted TD updates do not converge. When noise is added, TD diverges unless the discount is set low enough. When using fixed-horizon updates, convergence occurs, even in a high-noise, long-horizon setting. The shaded area indicates errors exceeding the error at initialization. Y-axis indicates $\gamma$ and labels indicate the noise level $\sigma$.

## 3 ERROR PROPAGATION IN BOOTSTRAPPED VALUE LEARNING

This section demonstrates that noisy function approximation, combined with bootstrapped TD updates, can lead to divergence in a simple environment, even with on-policy updates. This example shows the necessity for discounting when bootstrapping is used and suggests a relationship between the noise introduced by function approximation and the amount of discounting required. We use this example to justify the development of an algorithm that can learn value estimates without bootstrapping.

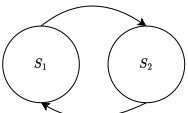

Figure 2: A two-state MDP. Rewards are always 0, the environment does not terminate, and there is only one action which takes the agent to the other state.

To demonstrate the problem, we construct a two-state fully connected MDP as shown in Figure 3. We then initialize a table containing state value estimates to 5 and update it according to value iteration (Sutton & Barto, 2018). Before each update, we perturb the table with noise sampled from $\mathcal{N}(\mu = 0, \sigma = \epsilon)$, which we call the noisy table setup. Since rewards are always 0 in this experiment, $V(s) = 0 \ \forall s \in S$. We consider an algorithm to have diverged if, after $100,000$ updates, the value table has an $L_1$ error of more than $10.0$.

We present the results in Section 3, under the noisy table setting for various values of $\gamma$. We find that TD updates diverged if the discount factor $\gamma$ was not set sufficiently low enough but that fixed-horizon updates converged even without discounting.

An intuitive explanation for this result is that errors can only propagate in one direction, from shorter horizons to longer ones, when performing fixed-horizon updates. This, coupled with the property that by definition $V_0 = 0$, provides a 'sink' for the errors. However, with TD learning, errors from one state propagate to the other during each update. Therefore, some dampening is required in the form of a discount factor $\gamma$.

This experiment underscores a significant limitation of TD learning-based algorithms. They are caught in a trade-off between substantial discounting and potential divergence issues in environments with high noise. This challenge is evident in the results from Agent57 on *Skiing*, which is both high-noise and long-horizon (Badia et al., 2020a). The workaround was extensive training. While fixed-horizon learning may circumvent this issue, its current constraint is the necessity to learn every horizon. Addressing the impracticality of learning all horizons comprehensively is the focus of our subsequent section.

## 4 Learning Long Horizons via Truncated Value Learning

In this section, we present our solution to long-horizon high-noise tasks in the form of Truncated Value Learning (TVL). TVL is based on the deep reinforcement learning (DRL) policy-gradient algorithm Proximal Policy Optimization (PPO) (Schulman et al., 2017). To which we apply an extension called Dual-Network Architecture (Aitchison & Sweetser, 2022), which, unlike PPO performs strongly on vision-based tasks, is fast to train, and includes a separate value network which may be required for the more complex value curve learning. To support efficient fixed-horizon learning on very long horizons, TVL introduces several important features:

1. Geometrically spaced value heads with linear interpolation allowing for arbitrarily long horizons using a fixed number of return estimates.
2. A sample-based return estimator (SRE) that enables efficient low-variance TD($\lambda$)-style return estimates in the fixed-horizon setting.
3. A method to apply fixed-horizon learning to the policy gradient setting, allowing for stochastic policies and more naturally suiting continuous control tasks.

Together these modifications allow the application of fixed-horizon learning to environments beyond the previous short horizons ($h = 64$), to very long horizons ($h = 27,000$) with a minimal impact on computational cost.

**Geometrically Spaced Value Heads.**

Managing individual models for every value function $V_{0..h_{\text{MAX}}}$ is infeasible. Instead, we consolidate these functions into a single value network that employs multiple value heads (Figure 4). Since the bulk of the computational effort lies in the shared encoder segment of the network, producing value estimates across numerous heads remains efficient. Nevertheless, generating return estimates to instruct these value heads poses a challenge.

To resolve this, we introduce a sampling and interpolation strategy. First, we learn value estimates for $K$ value heads at fixed locations spaced geometrically over the horizon range, enabling high precision for shorter horizons but less for longer ones. We, therefore, only need to update $K$ value heads during training rather than $h_{\text{MAX}}$. To produce estimates for horizons not learned, we use linear interpolation.

Formally, given a maximum horizon $h_{\text{MAX}}$ and a set number of horizons $K$ we generate an ordered set of horizon samples $\mathbf{H}$

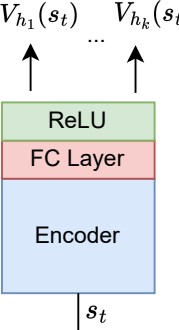

Figure 4: The architecture of the truncated value learning model in TVL. All value heads are estimated via different linear combinations of a shared feature representation.

$$\mathbf{H} = (\lfloor (h_{\text{MAX}})^{i/K} \rfloor)_{i \in 1..K} = (h_1, ..., h_K) \tag{8}$$

with duplicates permitted. Value estimates for horizons $h \in h_{\text{MAX}}$ can then be generated using linear interpolation (see Algorithm 3 in the Appendix). This allows TVL to learn only $K$ horizons, but output value estimates over an arbitrarily large range of horizons.

**Sample Based Return Estimator.** In the non fixed-horizon setting, an exponentially weighted sum of $n$-step estimates (TD($\lambda$)) can be calculated efficiently in constant time per value estimate regardless of the maximum $n$-step used (Sutton, 1988). However, this is not the case in the fixed-horizon setting. This is because in the traditional setting $n$-step estimates can be reused at different offsets by scaling by $\gamma^\delta$ where $\delta$ is the offset, but fixed-horizon estimates also require modifying the horizon $h$, and therefore need to be recomputed.

De Asis et al. (2020) provides an analogous algorithm (FHTD($\lambda$)) for the fixed-horizon setting. However, it requires generating $h_{\text{MAX}}$ value estimates for each of the $|\mathbf{H}|$ horizons learned and therefore scales poorly for very long-horizon problems.

To address this, we introduce a sample-based weighted return estimator (SRE) for generating return estimates via a weighting over n-step returns,

$$\text{SRE}_h^{\mathbf{W}}(s_t) := \mathbb{E}_{n \sim \mathbf{W}}[\text{NSTEP}_h^n(s_t)], \tag{9}$$

where $\mathbf{W}$ is a distribution over $\mathbb{Z}^+$. Algorithm 1 gives a method to generate SRE estimates. The advantage of SRE is that it supports any weighted mixture over $n$-step estimates yet only requires some constant $c$ value estimates. SRE can be used to recover estimates for standard return estimations, such as $n$-step and TD($\lambda$), as well as alternative distributions (see Appendix I).

---

**Algorithm 1** Sampled Return Estimator.

Input:
    $s \in S$, state for which value will be estimated.
    $h \in \mathbb{Z}^+$ horizon for which value will be estimated.
    $\text{NSTEP}_h^n(s)$ a $n$-step estimator than returns the $\text{n}^{th}$ $n$-step estimate of $V_h(s)$.
    $c \in \mathbb{Z}^+$ number of samples to use.
    $\mathbf{W}$ weights for each n-step estimate, with $w_i \geq 0 \; \forall \; i$ and $\sum_{i=1}^{|\mathbf{W}|} w_i = 1.0$.
Output:
    Estimate of $V_h(s)$ under under a weighted average of $n$-step estimators given by $W$.
$V_{est} \leftarrow 0$
**for** $i \in [1...c]$ **do**
    $x \sim \mathbf{W}$, where $p(x = i) = w_i$
    $V_{est} \leftarrow V_{est} + \text{NSTEP}_x(s, h)/c$
**end for**
**return** $V_{est}$

---

**Fixed-horizon Update Rule.** Fixed horizons can be learned in much the same way as traditional horizons by summing losses over the individual value heads

$$\mathcal{L}_t^{FH} = \sum_{h \in \mathbf{H}} \left[ \hat{\mathbb{E}}_t[(\hat{V}_h(s_t) - G_h^*(s_t))^2] \right] \tag{10}$$

where $G_h^*(s)$ is a return estimate for the policy starting from $s_t$ with a truncated horizon of $h$ (e.g. SRE), and $\hat{V}_h$ is the models value head corresponding to horizon $h$. Return estimates can also be improved by 'trimming' the $h$ used by $G_h^*$ back when $t + h$ exceeds $t_{\text{MAX}}$. We discuss this further in Appendix G.

**Generation of Advantages.** Advantage estimates $\hat{A}_t$ at time step $t$ in PPO are generated using general advantage estimation (GAE) (Schulman et al., 2015) which can be written as

$$\hat{A}_t^{\text{GAE}(\gamma,\lambda)} = \text{TD}^{(\gamma,\lambda)}(s_t) - \hat{V}^\gamma(s_t) \tag{11}$$

for which we substitute in our longest fixed-horizon approximation

$$\hat{A}_t^{\text{GAE\_FH}(\gamma,\lambda)} = \text{TD}_{h_{\text{MAX}}}^{(\gamma,\lambda)}(s_t) - \hat{V}_{h_{\text{MAX}}}^\gamma(s_t). \tag{12}$$

For $t + h_{\text{MAX}} \geq t_{\text{MAX}}$ this substitution is exact. Otherwise, for $r_t = 1.0 \; \forall \; t$, the error is upper-bounded by $\gamma^{h_{\text{MAX}}}/(1 - \gamma)$. Therefore, setting $h_{\text{MAX}} = \frac{3}{1-\gamma}$ (three 'effective horizons') limits the error to $< 5\%$, which we find to be sufficient.

Substituting this into the standard PPO clip loss, we have

$$\mathcal{L}_t^{\text{CLIP}} := \hat{\mathbb{E}}_t \left[ \min(\rho_t(\theta)\hat{A}_t^{\text{GAE\_FH}}, \text{clip}(\rho_t(\theta), 1 - \epsilon, 1 + \epsilon)\hat{A}_t) + c_{\text{eb}} \cdot \text{S}[\pi(s_t)] \right] \tag{13}$$

where S is the entropy in nats, $\rho_t$ is the ratio $\frac{\pi(a_t|s_t)}{\pi_{\text{old}}(a_t|s_t)}$ at time $t$, $\epsilon$ the clipping coefficient, and $c_{\text{eb}}$ the entropy bonus coefficient.

**Distillation.** The DNA algorithm includes a distillation phase, where knowledge from the value network is transferred to the policy network. Features learned regarding *when* the rewards arrived might be helpful to the policy network. However, attempting to learn the entire value curve and the policy might prove too difficult for the policy network. Therefore, we adapt the distillation process to select $c_{vh}$ value heads evenly from the available set as distillation targets while always including the final value head. Therefore, setting $c_{vh} = 1$ selects only the final value head, and $c_{vh} = k$ selects all value heads.

$$\mathcal{L}_t^D(\theta) := \sum_{h \in \mathbf{H}'} \left[ \hat{\mathbb{E}}_t [(V_\pi(s_t) - V_V(s_t))^2] \right] + \beta \cdot \hat{\mathbb{E}}_t [\mathrm{KL}(\pi_{\mathrm{old}}(\cdot|s_t), \pi(\cdot|s_t))] \quad (14)$$

where $\beta$ is the policy constraint coefficient, and $\pi_{\mathrm{old}}$ is a copy of the policy before the distillation update, and **H**' the subset of horizons to distil.

**Summary.** We have outlined how to efficiently generate value estimates along value curves, using interpolated value heads, and a sample-based return estimator. Together these changes allow TVL to efficiently generate TD($\lambda$)-style return estimates on long-horizon tasks. We formalize the TVL algorithm in Algorithm 2, while also providing source code with the supplementary material.[3]

---

**Algorithm 2** Truncated Value Learning (TVL)

---

1: **Input** $N \in \mathbb{Z}^+$ rollout horizon.
2: **Input** $A \in \mathbb{Z}^+$ number of parallel agents.
3: **Input H** The set of horizons to generate esimates for.
4: **Input** $\pi$ the initial policy.
5: **for** $t = 1$ to ... **do**
6:     **for** $a = 1$ to $A$ **do**
7:         Run policy $\pi$ in environment $a$ for $N$ time steps
8:     **end for**
9:     Compute $V_{\mathrm{targ}} \leftarrow G_h$ for each $h \in \mathbf{H}$
10:     Compute $\hat{A}_t^{GAE\_FH}$
11:     $\pi_{\mathrm{old}} \leftarrow \pi$
12:     **for** $i = 1$ to $E_\pi$ **do**
13:         Optimize $\mathcal{L}_t^{CLIP}$ wrt $\theta_\pi$
14:     **end for**
15:     **for** $i = 1$ to $E_V$ **do**
16:         Optimize $\mathcal{L}_t^{FH}$ wrt $\theta_V$
17:     **end for**
18:     $\pi_{\mathrm{old}} \leftarrow \pi$
19:     **for** $i = 1$ to $E_D$ **do**
20:         Optimize $\mathcal{L}_t^D$ wrt $\theta_\pi$
21:     **end for**
22: **end for**
23: **Output** $\pi$

---

## 5 EXPERIMENTAL RESULTS

Having introduced the TVL algorithm, we test our algorithm empirically. Given the high variance returns, we opted for *Procgen* (Cobbe et al., 2020) to test TVL's learning ability in a high-noise setting. Additionally, we chose Atari-5 (Aitchison et al., 2022) to gauge the agent's overall performance and selected the Atari game *Skiing* to evaluate its performance in long-horizon scenarios. Full training details are provided in Appendix F. We also provide supplementary results under the easy *Procgen* setting in Appendix E, and Mujoco results in Appendix D.

**High Noise Environments.** We found TVL to perform very well on the challenging Procgen benchmark. The results are summarized in Figure 5. Compared to PPG, TVL demonstrated the greatest

---

[3]Source-code will also be provided via GitHub on publication

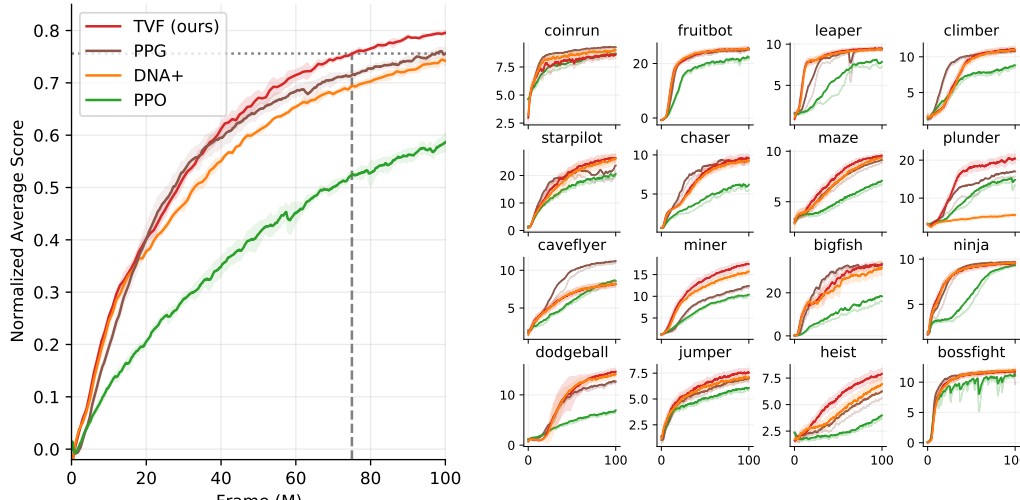

Figure 5: Results on the *Procgen* dataset over three seeds under the hard distribution setting. Left: Human normalized average score over all 16 games. Right: individual scores for each game. Results are averaged over three seeds and smoothed for readability. PPO and PPG results are taken from Cobbe et al. (2021). Shaded areas indicate 95% CI.

improvement on *Heist*, which requires complex long-term planning (unlocking a sequence of doors in a maze). It also performed well on *Maze*, which requires long-term planning. The largest difference found was on *Plunder*, where time penalties for shooting incorrect sprites led indirectly to reduced reward later on.

In terms of average normalized score, TVL outperformed all previous algorithms on this benchmark with a score of 0.796 (see Table 1), including the previous state-of-the-art phasic policy gradient (Cobbe et al., 2021) after 75 million frames. This is despite not requiring a large replay buffer and needing less time to train (see Appendix B).

| Algorithm | HNS | 95% CI |
|---|---|---|
| TVL (ours) | **0.796** | $0.792 - 0.799$ |
| PPG, (Cobbe et al., 2021) | 0.756 | $0.744 - 0.767$ |
| DNA, (Aitchison & Sweetser, 2022) | 0.742 | $0.733 - 0.751$ |
| PPO, (Cobbe et al., 2021) | 0.587 | $0.568 - 0.605$ |

Table 1: Average human normalized scores (HNS) for the Procgen benchmark under hard settings, including the 95-percent confidence interval.

**General Performance.** We assessed TVL's overarching performance using the Atari-5 dataset, which estimates the median score of Atari-57 based on just five games (Aitchison et al., 2022). Given that these games neither involve long horizons nor exhibit high noise, our aim was to determine if the emphasis on long-horizon learning would notably affect general performance.

We found TVL marginally under-performed DNA on the Atari-5 dataset, but the difference was not statistically significant. We also found no statistically significant difference in the five individual games evaluated. TVL demonstrated a considerable improvement over PPO, but less significant than that of DNA.

**Long Horizon.** We found that both TVL and DNA performed very well on the *Skiing* environment, with TVL achieving a time of 38.98 seconds (37.35-40.60 95% CI), and DNA a time of 40.61 seconds (38.47-42.75 95% CI) (Figure 6).[4] While TVL does not show a statistically significant improvement over DNA, it is the only algorithm tested that outperforms the previous best time of

---

[4]Scores in *Skiing* can be converted to (penalized) time in seconds by dividing by -100.

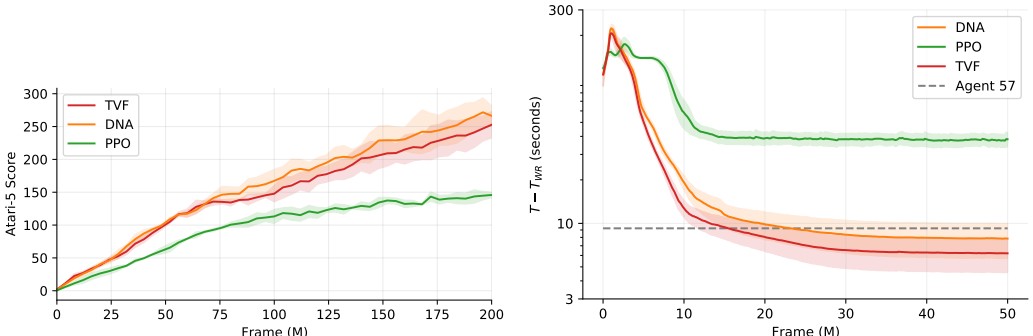

Figure 6: Left: Results on the Atari-5 benchmark averaged over three seeds. Shading indicates 95 % CI. Results smoothed for readability. Right: Results for the *Skiing* environment. The Y-axis indicates the number of seconds slower than the human world record, which is 32.78 seconds and is on a log scale. Shading indicates 95% CI over 16 seeds.

42.02 seconds by Agent-57 (Badia et al., 2020a) by a statistically significant margin.[5] We also note that 14 of the top 16 seeded runs were produced by TVL.

## 6    DISCUSSION

Our findings illustrate the potential of scaling fixed-horizon learning to extended horizons. While TVL exhibited superior performance in the long-horizon challenge *Skiing*, the margin of improvement was modest. Interestingly, its most remarkable performance was observed in the shorter-duration *Procgen* games. We speculate that this might be caused by the trimming approach not working as well in long-horizon settings as shorter ones. Exploring more sophisticated trimming techniques, such as time-until-termination prediction, could further enhance TVL's efficacy.

One limitation of our approach is that value heads in TVL are not independent due to feature sharing and interpolation. Because of this, errors on one head may propagate forward to longer-horizon heads. Despite this, TVL performs strongly in the high-noise *Procgen* setting. Also, when selecting test environments, we picked the longest-horizon problems we could find. However, finding tasks with exceedingly long horizons proved challenging. Given that many real-world problems, such as climate change, require judging the long-term implications of actions taken, it would be preferred if there existed established benchmarks involving longer time frames ($t_{\mathrm{MAX}} > 100,000$)

TVL demonstrated its most notable performance in the *Procgen* games, specifically in *Heist*, *Plunder*, and to a lesser extent, *Maze*. In contrast to other *Procgen* titles, these games place a greater emphasis on long-term planning. This suggests that TVL may be more adeptly bridge its immediate actions with far-reaching outcomes compared to other methods we evaluated. Further investigation into this potential strength would be beneficial.

## 7    CONCLUSIONS

In this paper, we have introduced fixed-horizon updates as a compelling alternative to TD learning. Our findings suggest that, when appropriately scaled, this method performs comparably on many tasks and occasionally outperforms alternative approaches. While we have not observed significant enhancements in long-horizon problems, our results in high-variance return environments are state-of-the-art. Given the theoretical benefits of fixed-horizon learning and its capability to simultaneously learn all discount functions, this research will likely encourage further exploration in the field.

---

[5]Go-Explore (Ecoffet et al., 2021) reports a time of 36.60 seconds. We do not compare against this result due to significant differences in the training approach (i.e. the exploitation of the perfect determinism of the environment and the modification of the environment's timeout.

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

## A   Truncated Value Interpolation Algorithm

This algorithm outlines a method for estimating the value of a state $s$ by using truncated value estimates at various horizons between $h_{\min}$ and $h_{\max}$, and then interpolating a value estimate for any arbitrary horizon $h$ within the range $[h_{\min}..h_{\max}]$ using linear interpolation. This approach decouples the number of value estimates generated from the maximum horizon $h_{\max}$, which is a key feature of TVL.

---

**Algorithm 3** Interpolated Value Estimate

---

Input:
  $s \in S$, state for which value will be estimated.
  $h \in \mathbb{Z}^+$ horizon for which value will be estimated.
  H, ordered list of K horizons.
  $\hat{V}$, list of K value estimates where $\hat{V}_i$ is an estimate for $V_{H_i}(s)$
  BISECT($a,x$) function that locates insertion point for $x$ in $a$ to maintain sorted order.
Output:
  Interpolated estimate of $V_h(s)$.

**if** h = 0 **then**
    **return**  0
**end if**

$r \leftarrow \text{BISECT}(H, h)$
$l \leftarrow r - 1$
$\Delta h \leftarrow H_r - H_l$
**if** $\Delta h = 0$ **then**
    **return**  $\hat{V}_l$
**end if**

$\alpha \leftarrow (h - H_l)/\Delta h$

**return**  $(1 - \alpha)V_{H_l} + \alpha V_{H_r}$

---

## B   Compute Efficiency of TVL

Implementing fixed-horizon learning poses a challenge in efficiently learning horizon estimates for problems with long horizons. In this section, we analyze the training time required to train TVL compared to PPO and DNA. TVL was trained on the Atari game *Name This Game*. The game was trained with a horizon limit of 30,000 and 4 return samples.

The training time required for TVL is similar to that of PPO as reported by Schulman et al. (2017). However, compared to RainbowDQN Hessel et al. (2018) TVL's training time is much faster. Despite learning value estimates for 128 different horizons, TVL only takes roughly twice as long to train as DNA. We observe that training time only begins to increase significantly when the product of the number of value heads and return samples exceeds 2,048.

Table 2: Approximate training times for the algorithms used in this paper.

| Algorithm | GPU hours per game |
|---|---|
| PPO (our settings) | 3 |
| DNA | 4 |
| PPO (Schulman et al. (2017) settings) | 7.5 |
| TVL | 9 |
| Rainbow DQN | 120[6] |

---

[6]When trained on an RTX 2080 using the code provided at `https://github.com/Kaixhin/Rainbow`.

Table 3: Impact of return samples ($c$) and number of value heads on training time.

| Value Heads | $c = 1$ | $c = 2$ | $c = 4$ | $c = 8$ |
|---|---|---|---|---|
| 64 | 9 hours | 9 hours | 9 hours | 9 hours |
| 128 | 9 hours | 9 hours | 9 hours | 9 hours |
| 256 | 9 hours | 9 hours | 9 hours | 10 hours |
| 512 | 9 hours | 9 hours | 10 hours | 12 hours |

## C   TRUNCATED RETURN ESTIMATORS

To effectively learn truncated value estimates $V_h$, obtaining accurate estimates of the return $G_h$ is crucial. While Monte-Carlo estimates are unbiased and simple to generate, they are plagued by high variance and are unsuitable for long episodes where estimates are required before episode termination. To address these issues, we present a set of return estimators in the fixed-horizon setting.

The Fixed-horizon setting introduces two challenges not present in traditional exponentially discounted value estimates. First, we need to estimate the value at multiple horizons, and second, the self-similar property of exponential discounting is absent, leading to less efficient TD($\lambda$) algorithms.

We review traditional return estimators that overlap with the Q-learning estimators developed by De Asis et al. (2020), as well as novel estimators that account for the specific needs and constraints of the fixed-horizon setting.

**Traditional Return Estimators Adapted for TVL.**

It is possible to modify the traditional return estimators TD, N-STEP, Monte Carlo (MC) and TD($\lambda$) to generate truncated return estimates with relative ease.

**One-Step Fixed-Horizon Targets.**

Return estimates for the truncated value can be learned in much the same way as a traditional value function using one-step updates as outlined by De Asis et al. (2020).

$$\text{TD}_h(s) := \text{NSTEP}_h^{(1)}(s) := \mathbb{E}_{\pi,\mu}\Big\{ r_t + \gamma V_{h-1}(s_{t+1}) | s_t = s \Big\} \tag{15}$$

$$\tag{16}$$

where $t$ is the current time step, $r_t$ is the next-step reward, and $\gamma \in \mathbb{R}$ the discount rate.

A drawback of using this return estimate is that any improvement made at the first horizon will require $h_{\text{MAX}}$ updates before it impacts the final horizon. This results in a slow propagation of important information, especially for longer horizons. To overcome this limitation, we can utilize $n$-step estimates.

**Monte Carlo Fixed Horizon Targets.** Monte Carlo estimates provide an unbiased estimate of the truncated return from a state but do so with high variance. We define Monte-Carlo return targets as

$$\text{MC}_h(s) := \mathbb{E}_{\pi,\mu}\Big\{ \sum_{i=0}^{h-1} \gamma^i r_{t+i} | s_t = s) \Big\} \tag{17}$$

where $r_{t > t_{\text{MAX}}} := 0$.

**n-Step Fixed-Horizon Targets.** Targets can also be estimated by considering not just the next-step reward, but a sequence of the next $n$ steps using

$$\text{NSTEP}_h^{(k)}(s) := \mathbb{E}_{\pi,\mu}\Big\{ \sum_{i=0}^{k-1} \gamma^i r_{t+i} + \gamma^k V_{h-k}(s_{t+k}) | s_t = s \Big\}$$

For $k \geq t_{\text{MAX}} - t$, $n$-step returns reduce to Monte-Carlo estimates since $V_{\leq 0} := 0$.

**TD($\lambda$) Fixed Horizon TD Targets.** TD($\lambda$) balances bias and variance in return estimation by using an exponentially weighted mixture of $n$-step return estimates. We can easily extend this to the fixed horizon setting as follows.

$$\text{TD}_h^{(\gamma,\lambda)}(s) := (1 - \lambda) \sum_{k=1}^{\infty} \lambda^{k-1} \text{NSTEP}_h^{(\gamma,k)}(\tau), \tag{18}$$

In the geometrically discounted setting, TD($\lambda$) has an efficient implementation that makes it no slower than a one-step estimate. However, all $h_{\text{MAX}}$ return estimations must be calculated in fixed horizons, which presents a challenge for long-horizon problems. To address this issue, we propose the following novel value estimators.

**Novel Value Estimators for TVL.**

After discussing the fundamental return estimates utilized in geometric horizon TD and introducing their fixed-horizon counterparts, we explore our novel return estimators that are more suited for fixed-horizon settings. These estimators aim to balance variance and bias trade-offs while remaining sub-linear in terms of the maximum horizon.

**Uniform Fixed-Horizon TD Targets.** We define the uniform return estimate as

$$\text{UNI}_h^N(s) := \frac{1}{N} \sum_{n=1}^{N} \text{NSTEP}_h^n(s) \tag{19}$$

which is a uniformly weighted average over $n$-step returns up to some step $N$. The reason for this particular formulation, is that if $N$ is set to the number of steps in an experience window, then transitions $(s_i, a_i, s_{i+1})$ all use $\text{NSTEP}_h^{N-i}$ as there target where $i$ is their position within the experience buffer.

Since $\text{NSTEP}_h^{N-i}(s_i)$ requires $V_{h-(N-i)}(s_{i+N-i})$ to be calculated, all estimates required are for $s_N$, that is, the last state. Therefore only one in every $N$ states requires horizon estimates to be generate.

If we set $N$ proportional to $h_{\text{MAX}}$, we have a constant number of return estimations independent of $h_{\text{MAX}}$. One downside to this style of return estimation is that all returns within a trajectory window use the same state for updating, which could cause issues.

## D    ADDITIONAL RESULTS ON MUJOCO

Given that TVL operates as a policy gradient algorithm, it is inherently tailored for continuous control problems, prompting us to present experimental results on MuJoCo (Todorov et al., 2012). While MuJoCo typically features low noise and short horizons, TVL's design is not expected to enhance performance in this context. Nevertheless, our goal is to ascertain if the introduction of learning the value curve has any adverse effects on the performance across these tasks.

As with the Atari and Procgen benchmarks, we also provided a time component to the model by appending $t/t_{\text{max}}$ to the state space. Because of this and other minor variations in hyperparameters, our results for PPO and DNA differ slightly from that of Aitchison & Sweetser (2022)

We found TVL to underperform DNA on this benchmark slightly. TVL improved upon DNA on one environment, *Walker2d*, and underperformed on two (*HalfCheetah*, and *Hopper*). All other environments showed no statistically significant difference over 30 seeds.

These results are within expectation, as TVL's primary advantage is the ability to generate long-horizon value estimates in noisy environments. The MuJoCo environments are limited to only 1000 time steps. Our agents used the standard $\gamma = 0.99$ discounting, giving an effective planning hori-

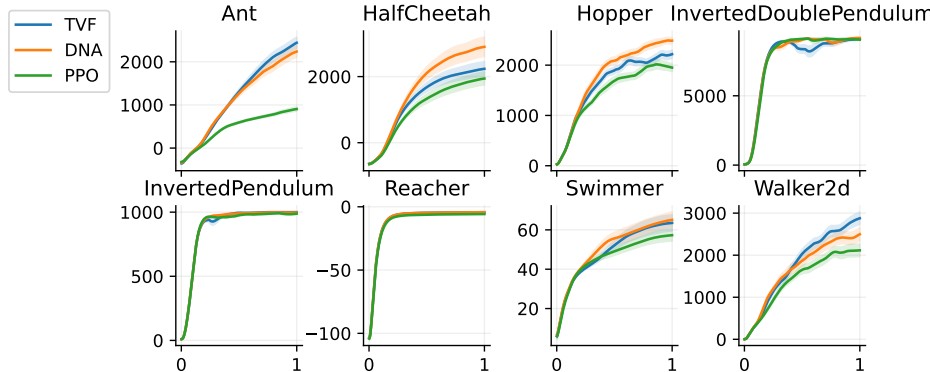

Figure 7: Results on the MuJoCo benchmark. Scores are averaged over 30 seeds, with one standard error shown shaded. All agents are trained for 1 million frames. Results are smoothed with an exponential moving average.

Table 4: Hyperparameters used for MuJoCo. Hyperparameters follow closely to that of Schulman et al. (2017); Aitchison & Sweetser (2022). Agents were trained for one million interactions. † Learning rate was annealed linearly to 0.0 over training.

| Setting | PPO | DNA | TVL |
|---|---|---|---|
| Entropy bonus ($c_{eb}$) | | 0.00 | |
| Rollout horizon (N) | | 2048 | |
| Parallel agents (A) | | 1 | |
| PPO epsilon $\epsilon$ | | 0.2 | |
| Discount gamma ($\gamma$) | | 0.99 | |
| Learning Rate | | $3.0 \times 10^{-4}$† | |
| Policy lambda ($\lambda_\pi$) | 0.95 | 0.9 | 0.9 |
| Value lambda ($\lambda_V$) | | 0.95 | |
| Policy epochs ($E_\pi/E_{ppo}$) | 10 | 10 | 10 |
| Value epochs ($E_V$) | - | 10 | 10 |
| Distil epochs ($E_D$) | - | 10 | 10 |
| Distil beta ($\beta$) | - | 1.0 | 1.0 |
| Policy mini-batch size | 64 | 64 | 64 |
| Value mini-batch size | - | 64 | 64 |
| Distil/Aux mini-batch size | - | 64 | 64 |
| Global gradient clipping | | 5.0 | |
| Embed time | | yes | |
| Distil max heads | - | - | All |
| TVL return samples | - | - | 8 |

zon of 100 time steps.[7]. Nonetheless, these results demonstrate TVL's retains it's ability to solve continuous control problems at a high level.

# E    ADDITIONAL RESULTS ON THE PROCGEN DATASET

Given our interest in evaluating TVL's performance in high-noise environments, in primary experiments evaluated on the hard *Procgen* settings. Nonetheless, several previous studies such as (Cobbe et al., 2020; Laskin et al., 2020; Raileanu et al., 2021) used the easy settings. For a comprehensive compassion with these works, we present results under the easy *Procgen* settings. Each model was

---

[7]We also tried higher values of $\gamma$ for both TVL and DNA, but found the standard 0.99 to be optimal from those tried [0.99, 0.999, 1.0]

trained for 25 million steps on the easy distribution and is evaluated using an average of 100 episodes at the end of training. TVL outperformed the other algorithms we compared against on 12 of the 16 games and achieved a PPO-normalized score of 164.6, outperforming the next best (DrAC) by 37%.

| Algorithm | PPO-Normalized Score |
|-----------|---------------------|
| PPO       | 100.0               |
| RAD       | 109.1               |
| DrAC      | 119.6               |
| TVL       | **164.6**           |

Table 5: Performance on Procgen under the easy settings, trained for 25 Million steps. PPO, RAD and DrAC results are taken from Raileanu et al. (2021). Scores are normalized by dividing by the PPO score for each game, multiplying by 100 and then averaging. TVL is averaged over three seeds, while PPO, RAD and DrAC are averaged over ten.

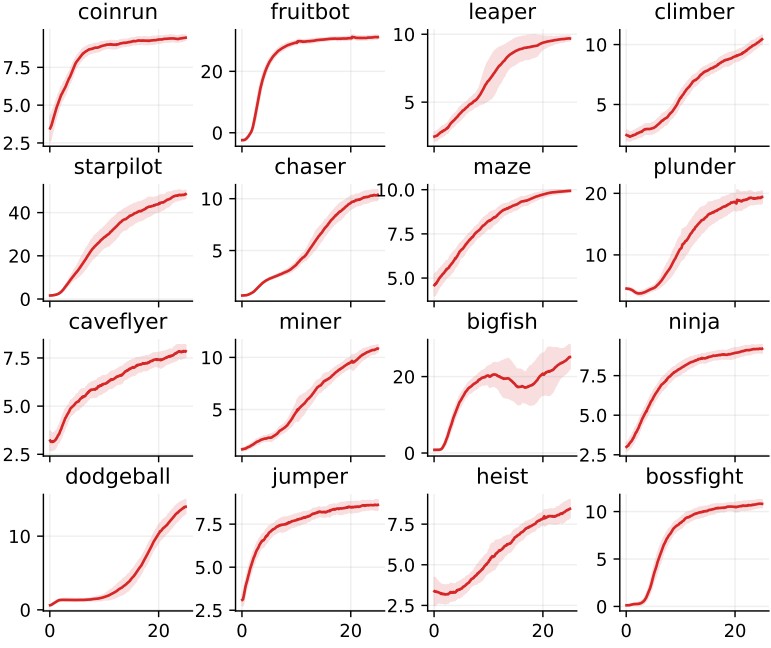

Figure 8: Individual training plots for TVL on each of the 16 games in the Procgen benchmark. Trained for 25 million steps under the easy settings. Shading indicates 95% CI over three seeds.

Table 6: Results for each game under the easy Procgen settings. PPO, RAD and DrAC results are taken from Raileanu et al. (2021). Bounds indicate two standard errors over three seeds for TVL and 10 seeds for the others.

| Game | PPO | RAD | DrAC | TVL |
|---|---|---|---|---|
| CoinRun | $9.3 \pm 0.3$ | $9.6 \pm 0.4$ | $\mathbf{9.7 \pm 0.2}$ | $9.5 \pm 0.1$ |
| StarPilot | $29.8 \pm 2.3$ | $36.5 \pm 3.9$ | $38.0 \pm 3.1$ | $\mathbf{51.5 \pm 0.6}$ |
| CaveFlyer | $6.8 \pm 0.6$ | $6.0 \pm 0.8$ | $\mathbf{8.2 \pm 0.7}$ | $7.6 \pm 0.4$ |
| DodgeBall | $4.2 \pm 0.5$ | $5.0 \pm 0.7$ | $7.5 \pm 1.0$ | $\mathbf{14.0 \pm 0.8}$ |
| FruitBot | $29.1 \pm 1.1$ | $26.1 \pm 3.0$ | $29.4 \pm 1.0$ | $\mathbf{31.3 \pm 0.6}$ |
| Chaser | $4.9 \pm 0.5$ | $6.4 \pm 1.0$ | $7.1 \pm 0.5$ | $\mathbf{10.5 \pm 0.2}$ |
| Miner | $12.2 \pm 0.3$ | $\mathbf{12.6 \pm 1.0}$ | $12.5 \pm 0.3$ | $10.5 \pm 0.4$ |
| Jumper | $8.3 \pm 0.4$ | $8.6 \pm 0.4$ | $\mathbf{9.1 \pm 0.4}$ | $8.8 \pm 0.7$ |
| Leaper | $5.5 \pm 0.4$ | $4.9 \pm 0.9$ | $5.0 \pm 0.7$ | $\mathbf{9.7 \pm 0.2}$ |
| Maze | $9.1 \pm 0.3$ | $8.4 \pm 0.7$ | $8.3 \pm 0.7$ | $\mathbf{10.0 \pm 0.1}$ |
| BigFish | $8.9 \pm 1.5$ | $13.2 \pm 2.8$ | $13.1 \pm 2.2$ | $\mathbf{25.1 \pm 3.6}$ |
| Heist | $7.1 \pm 0.5$ | $6.2 \pm 0.9$ | $6.8 \pm 0.7$ | $\mathbf{8.6 \pm 0.4}$ |
| Climber | $8.4 \pm 0.8$ | $9.3 \pm 1.1$ | $9.9 \pm 0.8$ | $\mathbf{11.0 \pm 0.4}$ |
| Plunder | $6.0 \pm 0.5$ | $8.4 \pm 1.5$ | $9.9 \pm 1.3$ | $\mathbf{19.6 \pm 1.7}$ |
| Ninja | $7.4 \pm 0.7$ | $8.9 \pm 0.9$ | $8.8 \pm 0.5$ | $\mathbf{9.4 \pm 0.1}$ |
| BossFight | $8.5 \pm 0.7$ | $8.1 \pm 1.1$ | $8.2 \pm 1.0$ | $\mathbf{10.7 \pm 0.8}$ |

## F  TRAINING DETAILS

**Changes from DNA.**

In our experiments we used settings similar to those used in Aitchison & Sweetser (2022). We use this section to note any important differences. Where we did make changes we applied those changes to all three algorithms evaluated.

**Entropy in Atari.** Our agent occasionally encountered situations where it became stuck, impacting the performance of PPO, DNA, and TVL. Specifically, it would either adhere to an under-performing policy or remain stationary in-game (e.g. holding a ball but not shooting). We identified two strategies to address these issues.

*1. The entropy bonus should be less on environments with less actions.*

We scaled the entropy bonus based on the number of valid actions as follows

$$e'_b = e_b \times \frac{\log(|A_{\text{target}}|)}{\log(|A|)} \tag{20}$$

where $|A_{\text{target}}|$ is the target action count (we set this to 18) and $|A|$ is the actual number of actions in the environment, with $|A| \geq 2$. This causes the entropy bonus for a uniform distribution to be the same regardless of the number of actions.

*2. The relative weighting of the entropy bonus is affected by advantage normalization.*

We modified the $\text{ADV}_\epsilon$ used during advantage normalization from $10^{-8}$ to $10^{-2}$. The reason for this is that the relative scale between the entropy bonus, and policy gradient is modified when advantages are normalized. This causes agents who receive no advantage (e.g. in environments where rewards are spare) so have an effective entropy bonus that is extremely high. By adjusting the $\text{ADV}_\epsilon$ parameter, the affect of this scaling is diminished in sparse reward environments.

**Procgen.** *Procgen* results were generated following the same procedure as Aitchison & Sweetser (2022), with the following changes.

- We apply distillation only on alternate updates.[8]

---

[8]We found this to be much faster, and produce similar results

- We increased the policy mini-batch size from 8k to 16k.

- We switched from RGB colorspace to YUV colorspace.

- We increased Adam $\beta_1$ from 0.9 to 0.95.

- We included the entropy modifications used in our Atari benchmarks.

Table 7: Hyperparameters used for ProcGen. PPG hyperparameters are taken from Cobbe et al. (2021).
† every other update.

| Setting | PPO | DNA | TVL | PPG |
|---|---|---|---|---|
| Color space | | YUV | | RGB |
| Advantage $\epsilon$ (ADV$_\epsilon$) | | 1e-2 | | 1e-8 |
| Entropy bonus ($c_{\mathrm{eb}}$) | | 0.01 | | |
| Rollout horizon (N) | | 256 | | |
| Parallel agents (A) | | 256 | | |
| PPO $\epsilon$ | | 0.2 | | |
| Discount gamma ($\gamma$) | | 0.999 | | |
| Learning Rate | | $5.0 \times 10^{-4}$ | | |
| Policy lambda ($\lambda_\pi$) | 0.95 | 0.8 | 0.8 | 0.95 |
| Value lambda ($\lambda_V$) | 0.95 | 0.9 | 0.9 | 0.95 |
| Policy epochs ($E_\pi/E_{\mathrm{ppo}}$) | 3 | 2 | 2 | 1 |
| Value epochs ($E_V$) | - | 1 | 1 | 1 |
| Distil/Aux epochs ($E_D$) | - | 2† | 2† | 6 |
| Distil/Aux beta ($\beta$) | - | 1.0 | 1.0 | 1.0 |
| Policy mini-batch size | 16,384 | 16,384 | 16,384 | 8,192 |
| Value mini-batch size | - | 2,048 | 2,048 | 8,192 |
| Distil/Aux mini-batch size | - | 512 | 512 | 4,096 |
| Global gradient clipping | 5.0 | 5.0 | 5.0 | off |
| TVL value heads | - | - | 128 | - |
| TVL return samples | - | - | 4 | - |
| TVL return distribution | - | - | exp | - |
| Adam Beta1 | 0.95 | 0.95 | 0.95 | 0.9 |

## G    TRIMMING AND VALUE PROPAGATION

A challenge in applying fixed-horizon learning to long-horizon tasks is the delayed propagation of improvements from short horizons to longer ones. In the most extreme case (when using NSTEP updates with $n = 1$) it would necessitate $h_{\mathrm{MAX}} - 1$ updates for an improvement at $V_1$ to propagate to $V_{h_{\mathrm{MAX}}}$. Since $\pi$ solely relies on $V_{h_{\mathrm{MAX}}}$, this might slow the agent's learning process.

Recognizing this, we exploit the following relationship

$$V_h(\cdot) = V_{h^+}(\cdot) \quad \forall\, h^+ \geq t_{\mathrm{MAX}} - t \tag{21}$$

Assuming $\hat{V}$ closely approximates $V$ it is pragmatic to replace longer horizons with shorter ones as the environment's time constrain nears, primarily because the latter are often learned earlier.[9]

We can, therefore, replace return estimations with either the shortest valid horizon $h_{\min} = \min(t_{\mathrm{MAX}} - t, \mathrm{h})$ giving

$$G_h^{TRIM}(s_t) = G_{h_{\min}}(s_t) \tag{22}$$

or alternatively with an average over all valid horizons shorter than the target horizon

---

[9]For example, at $t = 25$, and with $t_{\mathrm{MAX}} = 200$, $V_{>175}(\cdot) = V_{175}(\cdot)$, as termination beyond this horizon is assured.

$$\bar{G}_h^{TRIM}(s_t) = \frac{1}{1 + h - h_{\min}} \sum_{i=h_{\min}}^{h} G_i(s_t), \tag{23}$$

which may be preferred if return estimates have high variance.

## H  JUSTIFICATIONS FOR THE USE OF DISCOUNTING IN DECISION MAKING

The problems caused by discounting in human decision-making raise questions over why machines should discount at all. When justifying discounting in reinforcement learning, one or more of the following three reasons is generally given:

1. It ensures that return estimates are finite.

2. It makes sure that beneficial actions are not postponed indefinitely

3. It accounts for known (or unknown) hazards.

In a continuing non-episodic setting, all three justifications are well supported. For discussion see; returns are finite: Sutton & Barto (2018), postponing actions: Hutter (2004) (section 5.7), hazards: Sozou (1998). However, in an episodic environment where the time is limited by $t_{\mathrm{MAX}} \in \mathbb{Z}^+$, none of these reasons are well justified. If rewards are bounded by $R_{\max} \in \mathbb{R}$, then the returns will be bounded by $t_{\mathrm{MAX}} \cdot R_{\max}$. If an agent is aware of the current time step (required to make the environment Markovian (Pardo et al., 2018)), the agent will learn the optimal time to take any beneficial action. Finally, any environmental hazards will eventually be learned and compensated for automatically over time due to the ergodic nature of time-limited environments.

| Paper | Episodic | Time-limited | Discounting |
|---|---|---|---|
| Hessel et al. (2018) | Yes | Yes | $\gamma = 0.99$ |
| Mnih et al. (2015) | Yes | Yes | $\gamma = 0.99$ |
| Badia et al. (2020b) | Yes | Yes | $\gamma \in [0.99...0.997]$ |
| Badia et al. (2020a) | Yes | Yes | $\gamma \in [0.99...0.9999]$ |

Table 8: Summary of discounting used in popular RL papers.

This casts doubt on the need for any discounting in the time-limited episodic setting. Yet, findings from prominent RL papers in time-limited episodic domains consistently employ discounting, often over surprisingly short horizons (Table 8). Moreover, empirically, the discount rate is not an unimportant setting when tuning an algorithm but a critical hyperparameter. If the three reasons above do not hold, why not just set $\gamma = 1.0$ and learn the environment's unmodified rewards?

The reason given by many authors, including (Badia et al., 2020a; Schulman et al., 2015; Marbach & Tsitsiklis, 2003), is that discounting reduces the variance of the returns. That is to say, unknown far-future rewards are quite noisy, and making decisions using these estimates causes instability during training. By discounting, we reduce the weight of the long-term effect of a decision, thereby reducing the variance. While this allows the agent to learn more efficiently, it comes at the price of learning the *wrong problem*. Indeed, for any horizon, there exists an MDP where the optimal discounted policy is sub-optimal (even maximally so). This work's counterclaim is that discounting primarily functions in time-limited episodic environments to dampen error propagation caused by bootstrapping and that (high levels of) discounting is no longer required when bootstrapping is removed.

## I  ALTERNATIVE DISTRIBUTIONS FOR SRE

Traditionally, n-step returns are weighted exponentially for computational efficiency. In the fixed-horizon setting, recalculating these returns can be computationally intensive. To mitigate this, we employ a sample-based return estimator termed SRE. Notably, the SRE allows for a diverse range of n-step weights. We present several of these in Table 9. Furthermore, we emphasize the potential

of the harmonic series as an alternative to exponential weighting, offering the advantage of being parameter-free and not implying a specific timescale.

| Name | W |
|---|---|
| $n$-step | $w_i = 1$ if $i = n$ else $0$ |
| TD($\lambda$) | $w_i = \lambda^i$ |
| Uniform | $w_i = 1$ |
| Harmonic | $w_i = 1/i$ |

Table 9: Weights for various unnormalised distributions over N $n$-step estimates. Distributions can be normalized by dividing by $\Gamma = 1/\sum_{i=0}^{\text{N-1}} w_i$

.

## J    REDISCOUNTING

For simplicity, in our main study, we used a single discount $\gamma$ applied to both the return estimates used during value learning and advantage generation. However, this need not be the case. We can define $\gamma^{TVL}$ as the discount applied when learning value estimates and $\gamma^{PPO}$ the discount used when generating advantage estimates. Doing so provides several advantages.

1. Rediscounting, as explained below, involves averaging over multiple horizons, which might reduce variance.

2. The $\gamma^{\text{TVL}}$ could be set high (e.g. 1.0) while the $\gamma^{\text{PPO}}$ discount be could adjusted dynamically without destabilizing training by modifying the discount rate the value network is targeting, giving a dynamic potentially learned discount rate (for discussion see (François-Lavet et al., 2015; Wang et al., 2019)).

Using rediscounting can cause increased noise in value estimates if ratio $\phi_2(x)/\phi_1(x)$ is large for some $x$. When rediscounting, for example, from $\phi_1(x) = 0.99^x$ to $\phi_1(x) = 0.999^x$ at a horizon of 1000, any noise in the value estimation at $h = 1000$ will be magnified by $\approx 8517$ times. Consequently, it is often better to learn *longer* horizons and rediscount to *shorter* horizons than the other way around.

---

**Algorithm 4** General Rediscounting.

---

1: Procedure Rediscount($\phi_1$, $\phi_2$, $V_h^{\phi_1}(s)$) $\mapsto V_h^{\phi_2}(s)$
2: Input:
3:     Source discount function $\phi_1$ with support on $\mathbb{Z}^+$.
4:     Any target discount function $\phi_2$.
5:     $V_h^{\phi_1}(s)$, truncated values of state $s$ generated using discount function $\phi_1$ for $h \in 0...n$.
6: Returns:
7:     $V^{\phi_2} \in \mathbb{R}$, the truncated value of state $s$ under discount discount $\phi_2$ at horizon $n$.
8: $V^{\phi_2} \leftarrow 0$
9: **for** $h = 1 \rightarrow n$ **do**
10:     $r \leftarrow V_h(s) - V_{h-1}(s)$
11:     $V^{\phi_2} \leftarrow V^{\phi_2} + r \times \phi_2(h)/\phi_1(h)$
12: **end for**
13: **return** $V^{\phi_2}$

---

