# OpenReview forum: "Policy Gradient without Boostrapping via Truncated Value Learning"
_ICLR.cc/2024/Conference — Submitted to ICLR 2024_

### Official Review · Reviewer_tc1K · 2023-10-23

**Soundness:** 2 fair
**Presentation:** 2 fair
**Contribution:** 2 fair
**Rating:** 3
**Confidence:** 2

**Summary:**

The paper introduces a method that learns the value function with truncated horizon. This estimator is further combined with the PPO algorithm and is empirically tested on several Atari tasks. The experiment results show the advantage of the proposed algorithm.

**Strengths:**

The paper studies an important problem. The arguments are supported with sufficient examples and experiments. And the paper is presented in sufficient details and illustrations.

**Weaknesses:**

1. The motivation need to be further addressed. The authors demonstrate the advantage of horizon truncation in section 3, showing that untruncated TD update can lead to divergence. However, one weakness of this demonstration is the authors set h_max to be 1000, which is the effective horizon of $\gamma=0.999$. With this hard constraint, further increasing $\gamma$ has very little effect on the convergence result because they are "cut off" back to $0.999$ due to the constraint (as one can see from the figure, the slope is much smaller when $\gamma$ exceeds 0.999). In this case, the comparison for $\gamma > 0.999$ is not fair. On the other hand, when one looks the part of $\gamma < 0.999$, both curves show good convergence results, which makes the authors' claim less convincing.

The authors might strengthen their argument by adding the following results: 1. One would like to see what happens for a larger h_max value when testing the convergence result for a larger $\gamma$ value, for example, one can set h_max=10000 for $\gamma=0.9999$ so that no long term signals are killed. 2. Another concern is the introduction of h_max is exploiting the prior knowledge that there is no signal in the long term (because the reward is always 0 in this toy example.). If this is true, then setting an even smaller h_max values like 100 or 10 can lead to even better convergence result. The authors could also show the experiments for smaller h_max values to prove that this concern is wrong.

2. While the algorithm learns the h-horizon values for several horizons, when combined the value function with PPO, one only finds the appearance of the value function for h_max (for example, equation 12). This is somehow disappointing. It could be possible that the value estimations for various horizons are hidden somewhere in this expression like the TD updater. But if that is the case, the authors should address it in the main context as this is crucial for showing why their formulation is useful.

3. The use of notation is messy. In equation 3, while it's said to be n-step update, the corresponding notation is replaced by k in the equation. I never see a formal definition of NSTEP_h^n in the main article. The most similar notation is NSTEP_h^(k) in the appendix. The authors should clarify if they are pointing to the same thing, and why the bracket sometimes disappears, and why sometimes there is an additional $\gamma$ in the bracket but sometimes it also disappears. In the algorithm, the authors introduced another new notation NSTEP_x(s, h), I'm not sure if it has the same meaning of NSTEP_h^x(s). Overall, it could be much better if the authors could use consistent notations.

4. Part of the experiments didn't show convincing results. For both General Performance and Long Horizon, the algorithm doesn't significantly outperforms the baseline.

**Questions:**

1. What is the justification for using linear interpolation to estimate the value function of an unknown horizon $h$?

2. How did the authors choose the number of horizons K? If one uses a larger K, will one always expect a better performance so that the only constraint is computation resource? Or there is already some trade-off over the statistical performance so that increasing K can actually decrease the estimation accuracy / cumulative reward at some time point?

---

> ### Author Response · Authors · 2023-11-21
>
> Thank you for your insightful feedback and thoughtful comments. Please find below our responses to your queries:
>
> Point 1:
> Your point regarding the setting of h_max in the toy example is a good one. We set h_max to 1000 as this was the effective horizon of the first diverging TD case (gamma=0.999). However, it would have been better (for the reasons you have mentioned) to extend this beyond the effective horizon. We therefore have rerun our experiment with h_max = 10,000​ and found that Fixed Horizon (FH) updates did not diverge in either the 0.999 or 0.9999 gamma scenarios (for gamma=1.0, the error is unchanged from initialization). We look forward to updating our paper with these results, and they you very much for drawing our attention to this limitation. We believe this new data will more convincingly establish our claim regarding the divergence of TD in settings with high noise and high gamma, where FH does not. With the revised h_max, the impact of future rewards, even beyond this point, becomes negligible for a gamma of 0.999. This update should, hopefully, address your concerns by setting the horizon limit well above the effective horizon for our gamma values.
>
> We also agree that setting h_max low would cause the algorithm to perform well on this toy task (but not others) because it would essentially be applying heavy discounting. However, in this case, we are not concerned with the accuracy of the prediction so much as if the errors cause instability during training. Our intention is to show that even with very long horizons, TD still diverges but that this is not the case for FH.
>
> Point 2:
> We agree that exploring learned horizons could be advantageous. In earlier stages, we experimented with dynamically modifying the policy discount rate, effectively choosing shorter horizons from the learned value function. However, we found that opting for higher horizons was (almost) always beneficial. The crucial aspect of Trunched Value Learning (TVL) lies in its robust value learning, which remains stable even in noisy settings. After these estimates are learned, the final estimate is the one that matters, but the learning process necessitates the full range.
>
> Point 3
> We agree that the notation could be clearer. We changed this partway through and missed a few places where we needed to update it correctly. We would be more than happy to make the notation consistent and provide proper definitions for NSTEP_h^n (which is the n-step estimate truncated to a horizon of h). Our apologies for this omission.
>
> Point 4.
> We consider state-of-the-art results on the challenging Procgen benchmark a significant result. We were disappointed that TVL did not work well for long horizons, but we believe strongly that negative results should be published. This is an incremental step forward in an important area, and we show, at the very least, that the full horizon curve can be learned and that it improves performance in the challenging high-noise Procgen environment, which we believe would be of interest to the readers.
>
> Questions
> Why linear-interpolation? In earlier versions of the algorithm, we modelled the value curve directly, i.e. used a DNN to perform interpolation. This worked quite well but was overly complicated. We later discovered that linear interpolation worked just as well but was *much* simpler to implement, and we went for the simplest approach. Linear interpolation does have disadvantages. For example, if the environment was aperiodic, we might have an issue with rewards always appearing between the fixed head locations. We did not find this to be an issue in the environments tested, but we would be happy to write up this limitation if accepted.
>
> Number of heads? We found the algorithm to be robust to the parameter K. We did some initial hyperparameter searches and found 128 to be a good tradeoff between compute performance and the quality of the value estimates. We would be happy to include an ablation study on the values of K in an updated version of our paper for publication. There is a tradeoff in the quality of the value estimates (along the horizon curve) when using higher / lower K. However, this does not necessarily translate into large differences in performance (unless K is very low). The algorithm works quite well even with a coarse value curve representation.

---

### Official Review · Reviewer_ynUz · 2023-10-27

**Soundness:** 2 fair
**Presentation:** 3 good
**Contribution:** 1 poor
**Rating:** 5
**Confidence:** 3

**Summary:**

This paper proposed a deep PG method, called Truncated Value Learning (TVL), to learn rewards without discounting. In addition, this paper claimed bootstrap learning may degrade the performance in high-noise environments and introduced a bootstrap-free learning method. Some experimental results in  Procgen and Atari-5 seem to show an improved performance.

**Strengths:**

1. The paper is written concisely, clearly, and easily understandable.
2. The proposed method is relatively novel, and some experimental results show improved performance.

**Weaknesses:**

1. This paper lacks enough ablation experiments, making it difficult to see the role of each component clearly. For example, the author should have shown the benefits of TVL not using traditional discounting while learning rewards.
2. In addition, TVL has yet to be significantly proposed compared with the baseline algorithm on the long-horizon task, so it is difficult to judge whether the TVL method is effective on the long-horizon task. In addition, the author may be able to test it on more long-horizon tasks. Increase persuasiveness, such as pitfall, etc.

**Questions:**

1. Can the authors add some ablation experiments to supplement the paper?
2. There are many other long-horizon tasks in Atari, such as pitfall. Skiing alone is not convincing enough (and the performance in skiing does not seem to be significantly improved). Can the author add some results of other tasks? This might increase the convince of the proposed method.

---

> ### Author Response · Authors · 2023-11-21
>
> Thank you for your thorough review and valuable feedback.
>
> We agree with your suggestion that incorporating an ablation study would enhance the overall quality of our paper. We are planning to include an ablation study in the revised version of the paper, subject to its acceptance. This study will specifically examine the sensitivity of the newly introduced hyperparameters, and will add this study to the appendix. Our observations during the development of the Truncated Value Learning (TVL) algorithm indicated a robust performance across a wide range of hyperparameters, a detail that our current manuscript may not have adequately highlighted. We are looking forward to addressing this in the revised paper.
>
> In terms of comparing our algorithm to others in long-horizon scenarios, we believe our paper already presents these results. Specifically, we found that TVL does not surpass DNA in long-horizon tasks. We chose to use the Skiing game for evaluation over Pitfall, as Skiing presents a different challenge: rewards are deferred rather than sparse. This distinction allows for more informative rewards per episode compared to Pitfall, where most trajectories yield uninformative rewards. Where we did find improvement was in TVL's performance in a high-noise setting. Our interest in Skiing was due to it being the most challenging game for Agent57, which only solved it by increasing the discount to 0.9999. We believe this is a strong indicator that skiing is a good choice for long-horizon evaluation. While other games like Montezuma's Revenge and Pitfall also represent long-horizon challenges, they involve additional complexities related to hard exploration, an area our algorithm does not specifically aim to enhance.
>
> We appreciate your insightful review and hope our responses address your concerns. We look forward to improving our paper with the ablation study and other additional details in the revised version. Thank you once again for your constructive feedback.

---

### Official Review · Reviewer_wjX3 · 2023-10-27

**Soundness:** 3 good
**Presentation:** 3 good
**Contribution:** 2 fair
**Rating:** 5
**Confidence:** 3

**Summary:**

The paper presents a novel deep policy gradient algorithm, Truncated Value Learning (TVL), that can learn rewards discount-free while simultaneously learning value estimates for all summable discount functions. The main contribution of TVL is scaling the fixed-horizon long-horizon tasks with three ingredients: geometrically spaced value heads, sample-based return estimator, and fixed-horizon update rule.  The algorithm is tested empirically on the challenging high-noise Procgen benchmark and the long-horizon Atari game Skiing, showing state-of-the-art performance.

**Strengths:**

1. Effectively motivated to extend the learning horizon in fixed-horizon tasks.
2. The algorithm is rigorously evaluated on challenging benchmarks, specifically, noisy tasks where TD methods falter, and it surpasses previous state-of-the-art algorithms.

**Weaknesses:**

1. The novelty is under the bar for ICLR where this paper involves almost heuristic designs.
2. Long-horizon brings the higher sample inefficient for online algorithms. The training time is 3x higher than the PPO algorithm.
3. This paper involves multiple hyperparameters, e.g., $k, c_{vh}$, without sufficient ablation study.

Minors:
1. The lack of citation in the first time mentioning DNA algorithm.
2. The parameters in different networks should not be all $\theta$, try to use different notations for different networks.

**Questions:**

It is hard for me to follow to distillation part. For example, what is $c_{vh}$, and why we should use distillation? What is actually doing in Eq. 14?

---

> ### Author Response · Authors · 2023-11-21
>
> Thank you for your valuable observations regarding our paper, including the need to cite the DNA reference and update our notation for the policy and value network parameters theta. We appreciate these insights and will incorporate them into a revised version of our paper.
>
> Regarding the heuristic design of our solution, we acknowledge that it may not guarantee optimal results for long horizons. However, given the scarcity of algorithms capable of handling high discount factors (0.99997) and considering our state-of-the-art results on the Procgen benchmark, we believe our approach represents a significant advancement. We welcome and encourage future research that builds upon our work, aiming to enhance the algorithm with more robust guarantees. Our contribution lies in demonstrating the feasibility of such high-discount approaches, which, until now, was unexplored territory.
>
> Regarding the performance aspect of our algorithm, we acknowledge that it is three times slower than traditional approaches like PPO. However, this comparative decrease in speed represents a notable advancement compared to algorithms that scale linearly for large values of h. This sub-linear scalability is a significant attribute, demonstrating the robustness and potential of our method in handling very long time horizons, which is a crucial factor in complex environments and tasks.
>
> We recognize that the Truncated Value Learning (TVL) approach introduces additional hyperparameters. However, our findings suggest that the algorithm is quite robust to these parameters across the tested environments. Notably, TVL eliminates the need to fine-tune the gamma parameter per environment, as it performs consistently well with a high gamma across tasks. To further validate this, we are willing to include an ablation study in the appendix upon acceptance of the paper, demonstrating the sensitivity of the new hyperparameters introduced. In practice, we observed that TVL requires less tuning than algorithms like PPO or DNA, owing to its independence from environment-specific gamma adjustments.
>
> You have made a good point regarding the level of detail we included on the distillation process. We also plan to provide more detailed insights into the distillation process, recognizing its complexity and importance. To make our paper more self-contained, we will include a brief summary of the DNA distillation process.
>
> Thank you again for your constructive feedback, which will undoubtedly help in refining our paper and making it a more comprehensive and valuable contribution to the field.

---

### Official Review · Reviewer_Mwb3 · 2023-11-02

**Soundness:** 3 good
**Presentation:** 3 good
**Contribution:** 2 fair
**Rating:** 6
**Confidence:** 3

**Summary:**

The authors propose an algorithm for handling large horizons and undiscounted MDPs. They argue that discounting is for the purpose of reducing error propagation when bootstrapping. They claim that the proposed algorithm may work better in environments with high noise and high variance of returns.

**Strengths:**

The introduction is well-motivated and has a logical flow.

Proposed an algorithm that can learn discount-free rewards even for large horizons.

**Weaknesses:**

In the listed contributions what is the difference between points 1 vs 4 and 2 vs 3?
I think the authors should argue with the applications about the relevance of the study.

**Questions:**

Typo fig 3 caption, X-axis
Which step in the TVL algorithm is dampening the error propagation similar to discounting?

---

> ### Author Response · Authors · 2023-11-21
>
> Thank you for your insightful feedback. We acknowledge and appreciate your suggestion for greater clarity in describing the four contributions of our paper. To address this, we wish to clarify the distinction between our algorithm, highlighted as the first point, and the empirical results, presented as the fourth point. The algorithm is notable for being the first efficient fixed-horizon methodology, and the results are significant as they represent the first empirically results on long horizon problems, and are state-of-the-art on the challenging ProcGen benchmark. We would be happy to consider combining these into one contribution, however, as they are quite related.
>
> Regarding contributions two and three, the second contribution focuses on our novel approach to sample-based return estimation, particularly for calculating TD-lambda style returns. The third contribution, meanwhile, centers on the application of our value estimates in generating a policy gradient. This advancement is significant as it represents a new direction from previous works in the field, transitioning from Q-learning approaches towards policy gradient techniques.
>
> We are also grateful for your attention to the typo in figure-3. We will promptly revise the figure to accurately reflect that the x-axis represents the discount rate lambda, not the y-axis. Additionally, your comment regarding error dampening in the context of TD updates and TVL is well-taken. TD updates can propagate errors through loops, potentially leading to divergence if not adequately controlled. This is where TVL stands out, as it avoids loops in updates, thereby reducing the propensity for error propagation and diminishing the need for dampening. This aspect is indeed one of the major advantages of the TVL approach, and we will edit the contributions to make this more clear.
>
> We will incorporate these clarifications in the revised version of our paper to ensure a more comprehensive understanding of our contributions. Thank you again for your constructive feedback.

---

### Meta-Review · Area_Chair_8HrV · 2023-12-09

**Metareview:**

** Overview**:
The paper introduces a method that learns the value function with truncated horizon. This estimator is further combined with the PPO algorithm and is empirically tested on several Atari tasks. The experiment results show the advantage of the proposed algorithm.

** Review feebacks**:
- Reviewer Mwb3 rated the paper marginally above the acceptance threshold but highlighted the need for clarification on contributions and called for an ablation study.
- Reviewer wjX3, while appreciating the motivation and evaluation, pointed out heuristic designs and sample inefficiency, rating it marginally below the acceptance threshold.
- Reviewer ynUz noted the lack of robust ablation experiments and questioned the effectiveness of TVL on long-horizon tasks.
- Reviewer tc1K raised concerns about motivation, notation clarity, and experimental results, suggesting rejection.

**Justification For Why Not Higher Score:**

- The proposed algorithm, Truncated Value Learning (TVL), lacks sufficient novelty and does not meet the high innovation bar for ICLR.
- The paper's contributions in handling large horizons and undiscounted MDPs are overshadowed by the lack of clear differentiation of its primary features.
- The experimental results, while promising, do not significantly outperform existing baselines in long-horizon tasks.
- The use of multiple hyperparameters without comprehensive ablation studies weakens the paper's contribution.

**Justification For Why Not Lower Score:**

N/A

---

### Decision · Program_Chairs · 2024-01-16

Reject